# Integrative Utilization of Transcriptomics and Metabolomics Sheds Light on Disparate Growth Performance of Whiteleg Shrimp, *Litopenaeus vannamei*

**DOI:** 10.3390/ijms26073133

**Published:** 2025-03-28

**Authors:** Xin Zhang, Bo Ma, Pengying Li, Ting Chen, Chunhua Ren, Chaoqun Hu, Peng Luo

**Affiliations:** 1Sanya Institute of Ocean Eco-Environmental Engineering, Key Laboratory of Breeding Biotechnology and Sustainable Aquaculture, South China Sea Institute of Oceanology, Chinese Academy of Sciences, Guangzhou 510301, China; zhangxin@scsio.ac.cn (X.Z.); mabo20@mails.ucas.ac.cn (B.M.); chan1010@scsio.ac.cn (T.C.); rosemary166@sina.com (C.R.); 2University of Chinese Academy of Sciences, Beijing 101408, China; 3College of Fisheries and Life Science, Shanghai Ocean University, Shanghai 201306, China; lipengyingcafs@163.com

**Keywords:** *Litopenaeus vannamei*, growth trait, metabolomics, transcriptomics

## Abstract

*Litopenaeus vannamei* is a key economic species in aquaculture, yet the molecular mechanisms underlying its growth variability remain unclear. This study conducted transcriptomic and metabolomic analyses of fast-growing (NL) and slow-growing (NS) shrimp under identical conditions. A total of 1280 differentially expressed genes (DEGs) related to protein processing, ribosomes, and oxidative phosphorylation, along with 5297 differentially abundant metabolites (DMs) involved in arginine biosynthesis, amino acid metabolism, and pantothenate and CoA biosynthesis, were identified and analyzed. An integrative analysis revealed that the NL shrimp exhibited an enhanced retinol, glutathione, riboflavin, and purine metabolism, which implies a higher tolerance to environmental stress. In contrast, the NS shrimp showed increased fatty acid degradation and an accelerated TCA cycle. This suggests that NS shrimp might require a substantial amount of energy to cope with environmental changes, consequently resulting in increased energy expenditures. This study provides significant insights into the molecular mechanisms underlying the growth disparity in *L. vannamei*, offering valuable data for future research aimed at optimizing shrimp growth performance and enhancing aquaculture productivity.

## 1. Introduction

*Litopenaeus vannamei*, a species of significant economic importance, is renowned for its rapid growth and dominance of shrimp production worldwide, accounting for 82.7% of the total output [1,2]. Among the various traits influencing shrimp aquaculture, the growth rate is the most critical [3]. However, achieving synchronous growth in marine species remains a persistent challenge, leading to operational inefficiencies, extended farming cycles, and reduced commercial value due to size variability [4,5]. Similar growth disparities have been observed in other commercially important species, including *Crassostrea gigas*, *Pinctada maxima*, abalones, and *Ruditapes philippinarum* [6,7,8]. Understanding the molecular mechanisms underlying asynchronous growth is essential for improving the growth uniformity and optimizing the aquaculture productivity in *L. vannamei*.

Growth regulation is a complex process involving the interplay of numerous genes and metabolites [9]. Advances in molecular and sequencing technologies have enabled the large-scale identification of candidate genes associated with phenotypic traits [10]. Transcriptomics provides insights into gene expression patterns that directly influence phenotypes, whereas metabolomics offers a snapshot of tissue-specific metabolic profiles under specific conditions. Integrating these approaches provides a powerful framework for elucidating the key genes and metabolic pathways associated with the traits of interest. Previous studies on *L. vannamei* growth have focused primarily on environmental factors such as temperature, pH, salinity, and feed intake [11,12,13,14,15]. However, the molecular mechanisms driving asynchronous growth within the same culture environment remain unexplored.

Metabolomics is centered around the exploration of chemical processes linked to metabolites. Through metabolome analysis, it has been discovered that pearl oysters with varying growth rates display notable differences in amino acid biosynthesis and metabolism, along with glutathione metabolism [8]. In addition, metabolomics investigations have shown that metabolites associated with amino acid and fatty acid metabolism may play pivotal roles in the growth of *Macrobrachium rosenbergii* [16]. In largemouth bass, a metabolomics analysis has indicated that metabolites involved in carbohydrate metabolism can affect fish growth characteristics [17]. Integrated proteome and metabolome analyses revealed that, in *Crassostrea gigas*, the synthesis of proteins and polyunsaturated fatty acids ultimately contributes to an increase in biomass [18]. The combined transcriptomics and metabolomics analyses of *Stichopus monotuberculatus* and *Holothuria leucospilota* suggest that sea cucumbers with different growth traits possess distinct capabilities in lipid metabolism, protein synthesis [10], and oxidative stress regulation [6].

This study represents the first integrated transcriptomic and metabolomic analysis of *L. vannamei* to investigate the molecular basis of asynchronous growth. Using liquid chromatography–tandem mass spectrometry (LC–MS/MS) and transcriptomic sequencing, we compared the metabolite profiles and gene expression patterns between fast-growing (NL) and slow-growing (NS) shrimp. Correlation networks were constructed to identify linkages between metabolites and regulatory genes associated with growth performance. Our findings provide novel insights into the molecular mechanisms underlying asynchronous growth in *L. vannamei* and establish a theoretical foundation for enhancing shrimp aquaculture productivity.

## 2. Results

### 2.1. Significant Growth Performance Differences Between NL and NS

After 90 days of rearing, *L. vannamei* exhibited significant differences in growth performance (Figure 1A). The shrimp were categorized into two groups on the basis of body weight: the fast-growing group (NL, 7.59 ± 0.57 g) and the slow-growing group (NS, 1.58 ± 0.58 g). The average weight of the NL group was significantly greater than that of the NS group (*p* < 0.01) (Figure 1B), indicating a pronounced growth disparity under identical rearing conditions.

### 2.2. Metabolomic Analysis

To explore the metabolomic differences between the NL and NS groups, a principal component analysis (PCA) and partial least squares-discriminant analysis (PLS-DA) were conducted on the LC–MS/MS metabolomic data. PCA score plots in both positive and negative ion modes demonstrated a clear separation between the NL and NS groups (Figure 2A,B). This separation was further confirmed by orthogonal projections to latent structures discriminant analysis (OPLS-DA), which revealed distinct metabolite profiles between the two groups despite their shared environment (Figure 2C,D). The OPLS-DA model exhibited high predictability. In the positive and negative ion modes, the R^2^ values reached 0.627 and 0.657, respectively. Furthermore, when examining the regression line, its intersection points with the ordinate at the Q^2^ point were determined to be −0.14 and −0.16, as clearly shown in Figure 2E,F. These findings strongly attest to the model’s robustness and validate its appropriateness for subsequent in-depth analysis.

A total of 5297 differentially abundant metabolites (DMs) were identified, with 1753 metabolites upregulated and 3544 metabolites downregulated in the NL group compared with the NS group (Appendix A). A hierarchical clustering analysis revealed distinct metabolite profiles between the two groups in both positive (Figure 3A) and negative (Figure 3B) ion modes. A further analysis revealed 78 key metabolites (Appendix A), which were mapped to metabolic pathways via the Kyoto Encyclopedia of Genes and Genomes (KEGG) database (Figure 3C). The top enriched pathways included arginine biosynthesis, amino acid biosynthesis, histidine and purine-derived alkaloid biosynthesis, alanine–aspartate–glutamate metabolism, and pantothenate and CoA biosynthesis (Figure 3D). These pathways are likely critical in regulating the growth performance of *L. vannamei*.

### 2.3. Transcriptomic Analysis

A transcriptomic analysis revealed 1280 significantly differentially expressed genes (DEGs), with 748 genes upregulated and 532 genes downregulated in the NL group compared with the NS group (Appendix A, Figure 4A). The reliability of the transcriptomic data was validated via qPCR, which confirmed the expression patterns of ten randomly selected DEGs (Figure 4B). A Gene Ontology (GO) enrichment analysis revealed that the most enriched terms in the CC category were related to ribosomes, whereas the MF category was dominated by structural constituents of ribosomes. In the biological process (BP) category, the predominant terms included peptide metabolism, cellular amide metabolism, and translation (Figure 4C).

A KEGG pathway analysis of the DEGs highlighted the top 20 enriched pathways, including protein processing in the endoplasmic reticulum, ribosome function, protein export, thermogenesis, oxidative phosphorylation, actin cytoskeleton regulation, and the p53 signaling pathway (Figure 4D). These pathways are closely associated with cellular metabolism, energy production, and stress responses, suggesting their potential roles in growth regulation.

### 2.4. Integrative Analysis of Metabolomics and Transcriptomics

A Spearman correlation analysis was performed to integrate the transcriptomic and metabolomic data, revealing a strong correlation between the DEGs and DMs, and some of the results are presented in Figure 5A (Appendix A). The KEGG pathway analysis of the integrated data revealed several key pathways influencing growth performance, including fatty acid degradation, retinol metabolism, the TCA cycle, glutathione metabolism, riboflavin metabolism, and purine metabolism (Figure 5B).

In the fatty acid degradation pathway, genes such as *ACADVL* and *HADHA* were downregulated in the NL group, and the metabolite L-palmitoylcarnitine was less abundant in the NS group. With respect to retinol metabolism, the all-trans retinoic acid levels were lower in the NL group, whereas the *frmA* and *UGT* gene expression was upregulated. The TCA cycle pathway showed the upregulation of genes (*ACLY*, *PK*) and metabolites (isocitrate, oxaloacetate) in the NS group, suggesting potential energy metabolism dysregulation. Conversely, riboflavin and purine metabolism and glutathione metabolism pathways were upregulated in the NL group, with increased levels of key metabolites (riboflavin, hypoxanthine, γ-L-glutamyl-L-cysteine and glutathione) and genes (*ACP2*, *5′-nucleotidase*, *GCLM*, *GST* and *ANPEP*), indicating enhanced stress resistance and metabolic efficiency.

These findings collectively suggest that the NL group exhibited superior metabolic homeostasis and energy utilization, whereas the NS group may have experienced metabolic inefficiency and increased energy expenditures, contributing to its slower growth. This integrative analysis provides valuable insights into the molecular mechanisms underlying the growth disparities in *L. vannamei* and highlights potential targets for improving aquaculture productivity.

## 3. Discussion

### 3.1. Fatty Acid Degradation

The analysis revealed a significant enrichment of the fatty acid degradation pathway. Key genes involved in fatty acid degradation, such as *ACADVL* and *HADHA*, were downregulated in the fast-growing group. Fatty acid degradation is regulated primarily by long-chain fatty acids [19] and plays a critical role in energy acquisition through β-oxidation, which occurs mainly in peroxisomes [20] and mitochondria [21]. Three-hydroxyl-CoA dehydrogenase (HADHA) is a key enzyme in fatty acid oxidation and mitochondrial function, whereas very long-chain acyl-CoA dehydrogenase (ACADVL) is associated with mitochondrial reactive oxygen species (ROS) production and redox reactions [22]. The metabolomic analysis revealed that L-palmitoylcarnitine, a metabolite linked to immune responses [23], the stress response [24] and metabolic disorders [25], was less abundant in the slow-growth group. These findings suggest that the slow-growth group may undergo more frequent fatty acid oxidation, leading to greater energy consumption and potentially impaired metabolic homeostasis compared with the fast-growing group.

### 3.2. Retinol Metabolism

Retinol (vitamin A) plays a vital role in animal growth, development and cell differentiation [26]. It also influences ovarian maturation in shrimp [27]. Integrative transcriptomic and metabolomic analyses revealed significant enrichment of the retinol metabolism pathway in the fast-growing group. Although vitamin A itself is not the primary bioactive mediator, its derivatives, such as retinoate and 11-cis retinaldehyde, are critical for its functioning [28]. The fast-growing group presented lower levels of all-trans retinoic acid and upregulated expression of the *frmA* and *UGT* genes, suggesting enhanced energy metabolism through retinol pathway activation.

### 3.3. TCA Cycle

The tricarboxylic acid (TCA) cycle is central to the oxidation of sugars, proteins and lipids, serving as a metabolic hub for energy production [29]. Free fatty acids undergo β-oxidation to produce acetyl-CoA, which enters the TCA cycle [30]. Key enzymes such as ATP citrate lyase (ACLY) and pyruvate kinase (PK) are critical for energy metabolism [31]. Liu et al. reported that the expression of genes related to the TCA cycle was upregulated when *Platax teira* was subjected to cold stress [30]. In this study, genes and metabolites related to the TCA cycle, including *ACLY*, *PK*, isocitrate and oxaloacetate, were significantly upregulated in the slow-growth group. This upregulation may indicate disrupted energy metabolism and inefficient energy utilization in the slow-growth group, which is consistent with findings in other species under stress conditions.

### 3.4. Glutathione Metabolism

The glutathione metabolic pathway is essential for antioxidant defense, protecting organisms from oxidative damage [32]. Ma et al. reported that the glutathione metabolic pathway was significantly associated with the growth of sea cucumbers [10]. Glutathione-S-transferases (GSTs) play key roles in detoxification, oxidative stress prevention and intracellular transport [33]. In the slow-growth group, the levels of metabolites such as γ-L-glutamyl-L-cysteine and glutathione were reduced, and the expression of genes such as *GCLM*, *GST* and *ANPEP* was downregulated. These findings suggest that the slow-growth group has a diminished capacity to combat oxidative stress compared with the fast-growing group, potentially contributing to its impaired growth performance.

### 3.5. Riboflavin Metabolism and Purine Metabolism

Riboflavin (vitamin B2) is a precursor for flavin mononucleotide (FMN) and flavin adenine dinucleotide (FAD), which are essential for redox reactions in cellular metabolism [34]. It also protects tissues from oxidative stress by preventing lipid peroxidation and reducing damage from free radicals [35]. Riboflavin is synthesized from ribulose-5-phosphate (Ru5P) and guanosine-5′-triphosphate (GTP) [36], with contributions from the gut microbiota [37,38]. In the fast-growing group, the riboflavin levels were significantly greater, and the genes related to riboflavin and purine metabolism, such as *ACP2* and 5′-nucleotidase, were upregulated. This upregulation may increase stress resistance, improve feed utilization efficiency and promote muscle formation, ultimately supporting better growth performance [39].

## 4. Materials and Methods

### 4.1. Experimental Animals

A full-sib shrimp family was established in Maoming, China, following the methodology outlined in a previous study [40]. At approximately 90 days post-hatching, shrimp seedlings exhibiting significant size variation were selected for sampling. The NL group consisted of 18 large seedlings, whereas the NS group comprised 18 small seedlings. Muscle tissue was collected from each shrimp, with samples from three individuals within the same group pooled to form a single sample. This process yielded six samples for both the NL and NS groups.

### 4.2. LC–MS/MS and Metabolomic Data Analysis

Nontargeted metabolomics assays were conducted on the six samples from each group following established protocols. The process of LC–MS/MS mainly includes sample preparation, liquid chromatography separation, primary mass spectrometry (MS1), ion selection and fragmentation, secondary mass spectrometry (MS2) and data processing and analysis. The specific procedures followed the previously described methods [10].

The raw mass spectrometry data (.raw files) were processed via Compound Discoverer 3.1.0 (Thermo, Waltham, MA, USA) for peak extraction, retention time correction, adduct ion merging, gap filling, background peak labeling and metabolite identification. Each ion was characterized on the basis of retention time and *m*/*z* values, with recorded peak intensities, molecular weights, retention times, peak areas and identification results. The metabolites were annotated by matching the exact molecular mass, name and formula to the KEGG and HMDB databases, allowing for a mass difference of less than 10 ppm. The data analyses included a principal component analysis (PCA) and partial least squares-discriminant analysis (PLS-DA). The differential metabolite abundance between the NL and NS groups was assessed via Student’s *t* test, with the *p* values adjusted for the false discovery rate (FDR). Supervised PLS-DA with metaX and variable importance in projection (VIP) values was used to distinguish group variables. Differentially abundant metabolites (DMs) were identified on the basis of fold changes (≥2 or ≤0.5), *p* values (≤0.05) and VIP values (≥1). Further analyses of the DMs were conducted via MetaboAnalyst 5.0 and the KEGG database.

### 4.3. Transcriptomic Analysis for the Shriimp with Disparate Growth Performance

Total RNA was extracted from the shrimp’s muscle tissues via TRIzol reagent according to the manufacturer’s instructions. Poly(A) RNA was isolated from 5 µg of total RNA via poly-T oligo-attached magnetic beads, and cDNA libraries were prepared following the RNA-seq sample preparation kit protocol. The sequencing was performed as per the vendor’s recommendations. The raw reads from the transcriptome dataset were filtered to obtain clean data, and de novo assembly was carried out via Trinity 2.4.0. The gene expression levels were quantified via the transcripts per million (TPM) method. Differentially expressed genes (DEGs) were identified via the R package edgeR (3.20), with thresholds of |log2(fold change)| > 1 and a *p* value < 0.05. To validate the RNA-seq results, ten randomly selected DEGs were analyzed via qPCR in an independent shrimp population. The primer sequences are provided in Appendix A. Additionally, a functional enrichment analysis of the DEGs was performed via the KEGG and GO databases.

### 4.4. Integrative Metabolomics and Transcriptomics Analysis

The DEGs (*p* value < 0.05, |log2(fold change)| > 1) and DMs (*p* value < 0.05, |log2(fold change)| > 1, VIP > 1) were integrated for the correlation analysis. Pearson’s method was applied to calculate the correlation coefficients between the metabolomics and transcriptomics data via the R package. Heatmaps were generated to visualize the relationships between the DEGs and DMs.

## 5. Conclusions

The integrative analysis revealed that the fast-growth shrimp presented upregulated pathways linked to retinol, glutathione, riboflavin and purine metabolism, whereas the slow-growth shrimp presented increased fatty acid degradation and TCA cycle, which may lead to increased energy expenditures and impaired growth. Furthermore, this study highlights that riboflavin (vitamin B2) and L-γ-glutathione could serve as promising feed additives to increase shrimp growth and culture yields. These findings provide valuable insights into the molecular mechanisms underlying growth disparities in *L. vannamei* and highlight potential targets for improving aquaculture productivity.

## Figures and Tables

**Figure 1 ijms-26-03133-f001:**
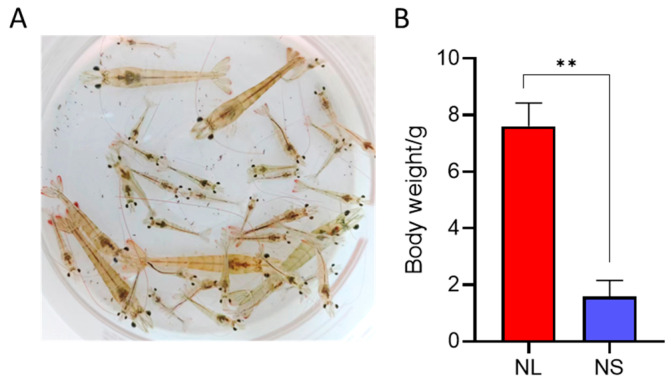
Growth performance of *Litopenaeus vannamei*. (**A**) Image showing varying growth rates in shrimp. (**B**) Statistical analysis of body weight (*n* = 5), with “**” indicating a significance level of *p* < 0.01.

**Figure 2 ijms-26-03133-f002:**
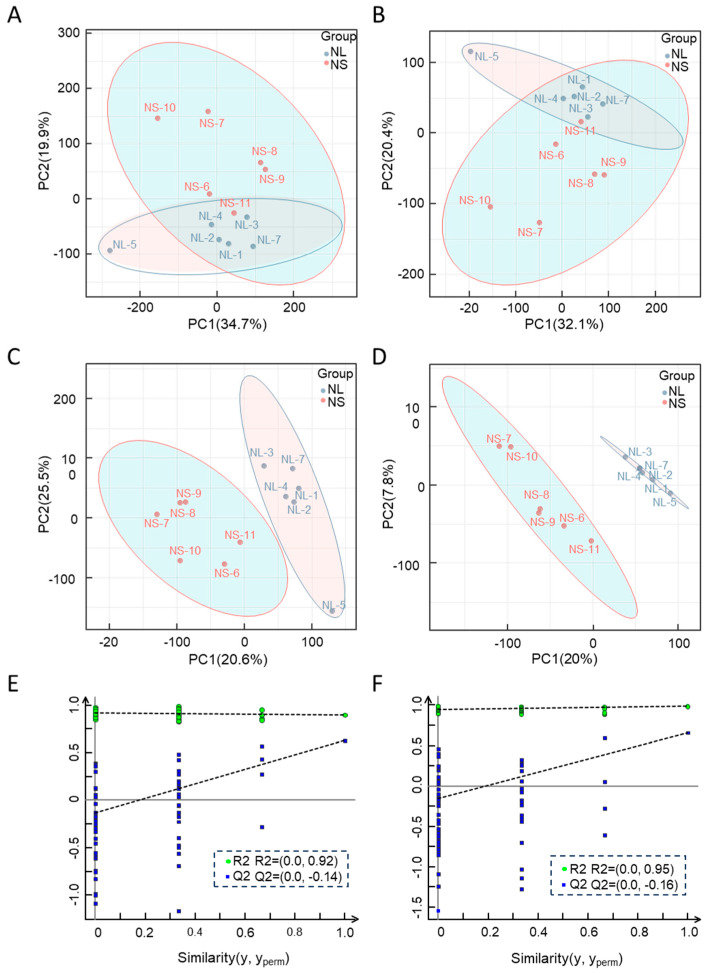
Metabolomics data quality assessment. (**A**) PCA score plot for positive ion mode samples. (**B**) PCA score plot for negative ion mode samples. (**C**) OPLS-DA score plot for positive ion mode. (**D**) OPLS-DA score plot for negative ion mode. Q^2^ and R^2^: coordinates where regression line intersects Y axis; same below. (**E**) OPLS-DA validation for positive ion mode. (**F**) OPLS-DA validation in negative ion mode.

**Figure 3 ijms-26-03133-f003:**
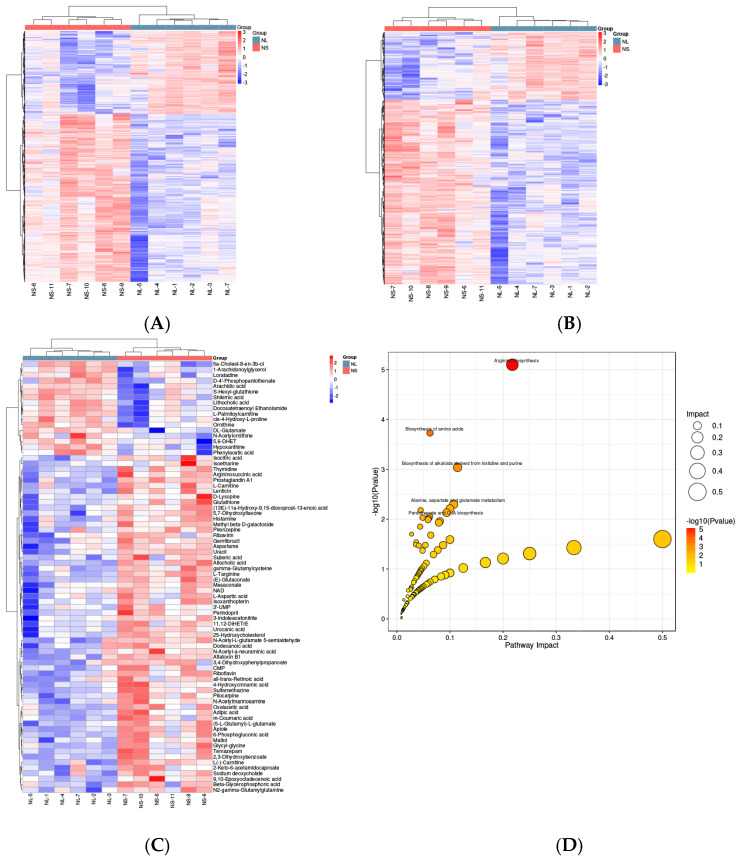
Metabolomic analysis of *Litopenaeus vannamei* in NS and NL groups. (**A**) Heatmap of hierarchical clustering of differentially abundant metabolites (DMs) between NS and NL groups. (**B**) KEGG pathway analysis of differentially expressed metabolites between NL and NS groups. (**C**) Hierarchical clustering heatmap of differentially abundant metabolites. (**D**) Bubble map of metabolic pathways associated with differentially abundant metabolites.

**Figure 4 ijms-26-03133-f004:**
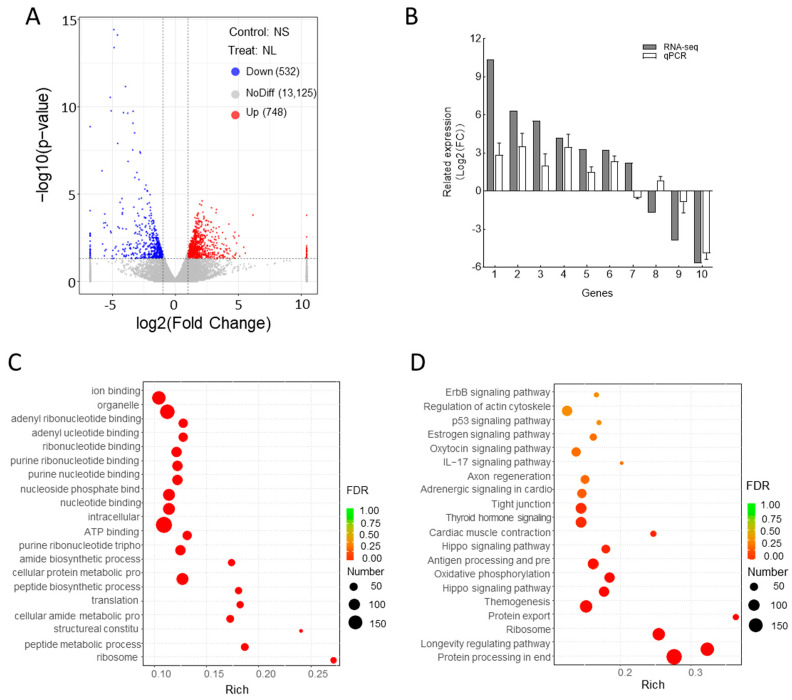
Transcriptomic analysis of *Litopenaeus vannamei* in NS and NL groups. (**A**) Volcano plot of differentially expressed genes (DEGs) between NS and NL groups. (**B**) Validation of RNA-Seq gene expression data, presented as means ± SDs (*n* = 3). (**C**) Gene Ontology (GO) functional enrichment of DEGs. (**D**) KEGG pathway enrichment of DEGs.

**Figure 5 ijms-26-03133-f005:**
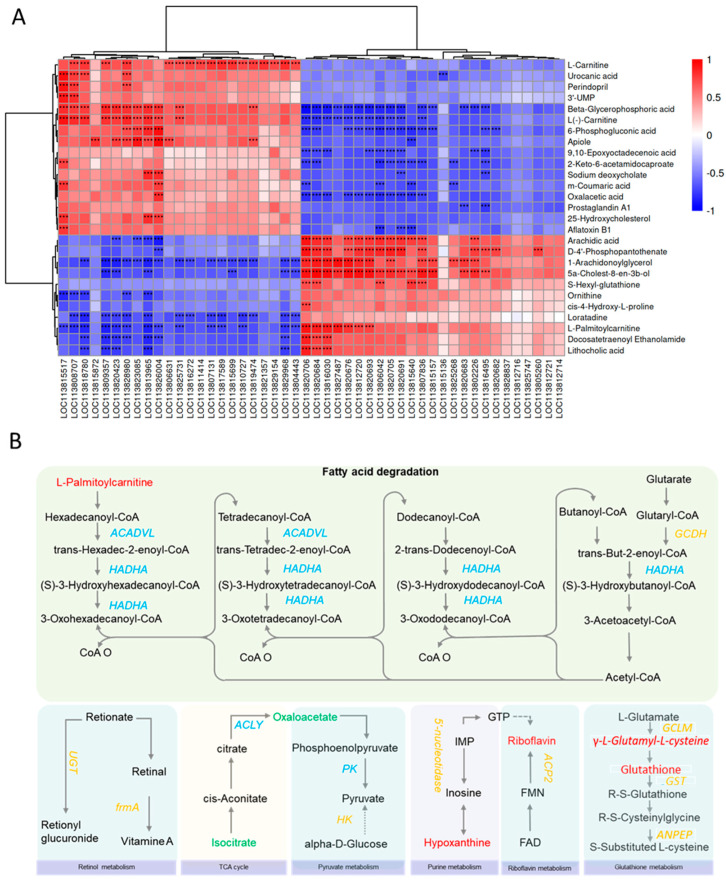
A correlation analysis between the transcriptome and metabolome of *Litopenaeus vannamei*. (**A**) A heatmap showing the gene–metabolite correlations. The columns represent genes, and the rows represent metabolites. Red indicates positive correlations, and blue indicates negative correlations. “***” denotes *p* < 0.001. (**B**) An integrated metabolic network map of the DEGs and metabolites between the NS and NL groups. The DEGs are italicized, with orange (upregulated) and blue (downregulated) in the NL group. The DMs are in Romania, with red (upregulated) and green (downregulated) in the NL group.

## Data Availability

The data presented in this study are available in the article.

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
