# Peer review of "Integrative Utilization of Transcriptomics and Metabolomics Sheds Light on Disparate Growth Performance of Whiteleg Shrimp, Litopenaeus vannamei"

_ijms, 2025, doi:10.3390/ijms26073133_

Round 1
Reviewer 1 Report
Comments and Suggestions for Authors
This paper conducts a comparative analysis of transcriptomics and metabolomics data between fast-growing and slow-growing individuals of cultured white-leg shrimp (Litopenaeus vannamei). Considering the limited knowledge available on shrimp transcriptomics and metabolomics, I am convinced that the content of this paper is of great significance. Overall, the manuscript is well-organized, and the results are clearly presented. The issues are listed as follows:
- Whether the observed transcriptomic and metabolomic differences between the two groups represent the characteristics of asynchronous growth, or represents the ontogenetic changes specific to this shrimp species? Could you offer an explanation?
- Were the shrimp in the NL and NS groups cultured under the same conditions? What were the sources of environmental stresses for the NS group during the culture?
- There are numerous abbreviations in this article. It is recommended that the authors provide a unified illustration of these abbreviations (e.g., on Lines 128, 131, 132). Please review the entire manuscript.
- Line 68: “growth rates” is inappropriate. “Body size” is more accuracy.
- Line 122: The subtitle of “Transcriptomic analysis” is incorrect. Please correct it.
- Line 122: The results of the integrative analysis for “Glutathione metabolism” should be included.
- Line 186: The result does not match Fig 5B. Please check and revise.
Comments on the Quality of English LanguageThe English could be improved to more clearly express the research.
Author Response
Comment 1: Whether the observed transcriptomic and metabolomic differences between the two groups represent the characteristics of asynchronous growth, or represents the ontogenetic changes specific to this shrimp species? Could you offer an explanation?
Response 1: In partnership with shrimp aquaculture stakeholders, we have established a meticulously controlled breeding system. By using gametes from a single mating pair of one female and one male penaeid shrimp, we successfully generated a genetically homogenous shrimp population. The fertilized eggs were methodically collected and then cultivated in ponds under standardized conditions, where every environmental factor was strictly regulated. This well-considered experimental design effectively minimized both genetic variability and environmental fluctuations, thus creating an ideal model system to explore the nongenetic factors contributing to growth variation. Although crustacean embryogenesis has limited morphological differentiation, significant growth disparities emerged during the 90-day cultivation period. This occurred because the shrimp shared the same genetic background and were exposed to consistent environmental conditions. To gain a deeper understanding of the molecular mechanisms underlying this growth heterogeneity, we adopted a comprehensive comparative multiomics approach. At the harvest stage, we employed extreme phenotype sampling. We carefully selected two cohorts of shrimp with distinct size differences, representing the upper and lower deciles of the growth distribution (fast-growing and growth-retarding individuals, respectively). These cohorts were then subjected to integrated transcriptomic and metabolomic profiling. This experimental design was crucial because it ensured that the observed molecular differences were specifically related to growth performance rather than being confounded by genetic or environmental factors.
Comment 2: Were the shrimp in the NL and NS groups cultured under the same conditions? What were the sources of environmental stresses for the NS group during the culture?
Response 2: The shrimp in both the NL and NS groups were progenies from the same pair of parent shrimp. After 90 days of artificial cultivation, these shrimp fry displayed asynchronous growth patterns. Environmental stressors could stem from fluctuations in water temperature, variations in pH levels, and other factors. We hypothesize that the NS group might be more susceptible to these environmental alterations.
Comment 3: There are numerous abbreviations in this article. It is recommended that the authors provide a unified illustration of these abbreviations (e.g., on Lines 128, 131, 132). Please review the entire manuscript.
Response 3: Thank you for your suggestion. We have revised this in the revised manuscript.
Comment 4: Line 68: “growth rates” is inappropriate. “Body size” is more accuracy.
Response 4: Thank you for your valuable suggestions. Acceptations and changes have been made.
Comment 5: Line 122: The subtitle of “Transcriptomic analysis” is incorrect. Please correct it.
Response 5: Thank you for your valuable suggestions. Acceptations and changes have been made.
Comment 6 Line 122: The results of the integrative analysis for “Glutathione metabolism” should be included.
Response 6: We apologize for the carelessness. We have modified this part in the revised manuscript.
Comment 7: Line 186: The result does not match Fig 5B. Please check and revise.
Response 7: Sorry for the carelessness. We have modified these issues carefully to improve the clarity.
In addition, we revised the introduction and made several minor mistakes.
Reviewer 2 Report
Comments and Suggestions for Authors
The growth disparities of White-leg shrimp, Litopenaeus vannamei in the same feeding condition may be caused by many factors. So here are some questions you need to answer.
1.Please provide the evidence that the NS shrimp may struggle to maintain metabolic homeostasis. Do the authors mean that there is metabolic disorder in the NS group?
2.Similar growth disparities have been observed in other commercially important species, including Crassostrea gigas, Pinctada maxima, abalones, and Ruditapes philippinarum. In the introduction section, cite more literature exploring the growth disparities of aquatic animals, especially using metabolomic technique.
3.Figure 3. Metabolomic analysis of Litopenaeus vannamei in the NS and NL groups . The figure information is incomplete, please supplement and improve it.
4.In the section 4.1. Experimental animals. The growth rate of shrimp will also vary depending on different farming modes. Provide information on the shrimp farming mode used in this study. It would be best if you could provide a picture.
5.The reference format cannot meet the publishing format requirements.
Author Response
Comment 1: Please provide the evidence that the NS shrimp may struggle to maintain metabolic homeostasis. Do the authors mean that there is metabolic disorder in the NS group?
Response: The shrimp were cultured in the pool for 90 days. During the entire culture period, distinct metabolic patterns were noted between the NS group and the NL group. In the NS group, the shrimp presented increased fatty acid degradation, retinol metabolism, and the tricarboxylic acid (TCA) cycle. These metabolic pathways are involved mainly in energy production. In contrast, the shrimp in the NS group presented decreased activities in glutathione metabolism, riboflavin metabolism, and purine metabolism, which are associated with stress protection mechanisms. These findings indicate that the shrimp in the NS group probably consumed a large amount of energy during their daily metabolic processes rather than experiencing metabolic disorders. Essentially, the shrimp in the NS group might be more vulnerable to the impact of environmental factors. In contrast, the shrimp in the NL group might have exhibited an active response to environmental stress. In the context of healthy shrimp breeding, a stable breeding water environment, a certain level of tolerance to water-environment changes, or an external environment that does not trigger a stress response is conducive to the growth of shrimp. The more in-depth regulatory mechanisms are likely to become the focus of future research. Moreover, I also refined the language in the revised manuscript.
Comment 2: Similar growth disparities have been observed in other commercially important species, including Crassostrea gigas, Pinctada maxima, abalones, and Ruditapes philippinarum. In the introduction section, cite more literature exploring the growth disparities of aquatic animals, especially using metabolomic technique.
Response 2: Thank you for your valuable suggestions. We have refined the introduction section to provide a more comprehensive background for this research. In the revised manuscript, we have incorporated relevant studies on metabolomic analysis conducted on several marine species, including the pearl oyster Pinctada maxima, the Pacific oyster Crassostrea gigas, abalones, and the Manila clam Ruditapes philippinarum.
Comment 3: Figure 3. Metabolomic analysis of Litopenaeus vannamei in the NS and NL groups. The figure information is incomplete, please supplement and improve it.
Response 3: Thank you for your valuable suggestion. The legends of Fig. 3C, D were added to the revised manuscript to improve the clarity.
Comment 4: In the section 4.1. Experimental animals. The growth rate of shrimp also varies depending on different farming modes. Provide information on the shrimp farming mode used in this study. It would be best if you could provide a picture.
Response 4: Undoubtedly, the growth-associated phenotypic traits of Litopenaeus vannamei are significantly influenced by many factors throughout the aquaculture process. In this study, to minimize the potential interference of external factors on growth phenotypes, we initially performed comprehensive transcriptomic and metabolomic analyses on individuals from full-sibling families. This methodological approach effectively eliminates the confounding effects of genetic disparities. Moreover, during the entire cultivation period, all shrimp from the same family were reared in a single culture pool. This experimental setup allowed for precise control of various environmental and nutritional factors, such as water quality parameters (including temperature, salinity, and pH) and feed composition, which could affect the growth performance of L. vannamei. All these details have been incorporated into the revised manuscript. Notably, the shrimp larvae initially hatched in a tank and then transferred to a culture pool for 90 days of cultivation. Figure 1 in the manuscript depicts the transfer of the larvae to the pool, whereas Figure 2 shows the larvae with different growth rates. The figures provided in the manuscript illustrate shrimp with distinct growth performances after the 90-day culture period. I hope this explanation is satisfactory.
Comment 5: The reference format cannot meet the publishing format requirements.
Response 5: Thank you for your valuable suggestions. We have revised the reference format to meet the publishing format.
In addition, we revised several minor mistakes.
Reviewer 3 Report
Comments and Suggestions for Authors
The manuscript by Zhang and colleagues studied the molecular mechanism underlying fast-growing (NL) and slow-growing (NS) white-leg shrimps (L. vannamei) in controlled conditions. Using transcriptomics and metabolomics, they found distinct genes and pathways between the two growth patterns, which include upregulated retinol, riboflavin, and purine metabolism in NL shrimps, whereas higher fatty acid degradation, the TCA cycle and energy expenditure in NS shrimps. They concluded that the metabolic mechanism can be used to enhance aquaculture productivity.
Comments and suggestions:
1. Fig 1B, it would be better to give the sample size in the legend.
2. Line 79-80, “Permutation tests yielded Q² values below zero (-0.14 for positive ion mode and -0.16 for negative ion mode), confirming the model's robustness…” If I’m not mistaken, for cross-validation, a negative Q^2 should indicate the model does not have predictive relevance while a Q^2 > 0.5 should indicate the model has a decent predictability. More clarification is needed here.
3. Fig 3. First, the legend misses panel C and D. Second, panel D, how is the “pathway impact” calculated? Why do the dots form a curve on the lower part of the figure?
4. Line 101 and 122, section 2.3 and 2.4 have identical titles. Do the authors mean “integrative analysis” for 2.4?
5. Line 144, panel A, x- and y-axis titles are needed. Did the authors perform multiple correction?
6. Section 2.4, could the authors make a Venn diagram to show the overlap and difference between the transcriptomic profile and the metabolomic profiles? That will be visually easier than the heatmap (Fig 5A), from which no particular genes/metabolites can be seen.
Author Response
Comment 1: Fig 1B, it would be better to give the sample size in the legend.
Response 1: Thank you for your suggestion. Five shrimp in each group were weighed, and changes have been made in the revised manuscript.
Comment 2: Line 79-80, “Permutation tests yielded Q² values below zero (-0.14 for positive ion mode and -0.16 for negative ion mode), confirming the model's robustness…” If I’m not mistaken, for cross-validation, a negative Q^2 should indicate the model does not have predictive relevance while a Q^2 > 0.5 should indicate the model has a decent predictability. More clarification is needed here.
Response: I apologize for my oversight. In the two modes, the intersection points of the regression line with the ordinate at the Q2 point are -0.14 and -0.16, respectively. Since both values are less than zero, the results are reliable. Moreover, the predicted parameters of the model are 0.627 and 0.657, respectively (Q2 > 0.5). We have revised the data in this graph and made corresponding adjustments to the revised manuscript.
Comment 3: First, the legend misses panel C and D. Second, panel D, how is the “pathway impact” calculated? Why do the dots form a curve on the lower part of the figure?
Response 3: I apologize for my oversight. In the revised manuscript, the legends for panels C and D have been provided. With respect to the calculation of pathway impact, we conducted differentially abundant metabolite pathway analysis via the MetPA database, which is based mainly on KEGG metabolic pathways. This analysis aimed to pinpoint potentially biologically perturbed metabolic pathways through a combination of metabolic pathway enrichment and topological analysis. Specifically, we used the MetPA database to examine the relevant metabolic pathways of differentially abundant metabolites between the two groups. For the data analysis, we employed the hypergeometric test as our algorithm. To evaluate the pathway topological structure, we utilized the relative betweenness centrality method. On the basis of the results obtained from the MetPA analysis, we applied dimensionality reduction algorithms to calculate the relative response values of the identified metabolites within the metabolic pathways. These values were then used to compute the correlation coefficients between different metabolic pathways. We subsequently constructed metabolic pathway interaction network diagrams. Finally, to present the results in a more intuitive manner, we visualized them as bubble plots on the basis of the calculated data. We sincerely hope that this explanation addresses your concerns.
Comment 4: Line 101 and 122, section 2.3 and 2.4 have identical titles. Do the authors mean “integrative analysis” for 2.4?
Response 4: We apologize for the carelessness. We have revised the title to “Integrative Analysis of Metabolomics and Transcriptomics”.
Comment 5: Line 144, panel A, x- and y-axis titles are needed. Did the authors perform multiple correction?
Response 5: Thank you for your suggestion. A new figure is provided, the x-axis represents the functional genes, and the y-axis represents the metabolites. This information was added to the legend of Figure 5A in the original manuscript. Additionally, the correlations between the genes and the metabolites were subjected to multiple correction, and the
Comment 6: Section 2.4, could the authors make a Venn diagram to show the overlap and difference between the transcriptomic profile and the metabolomic profiles? That will be visually easier than the heatmap (Fig. 5A), from which no particular genes/metabolites can be seen.
Response 6: Thank you for your valuable suggestion. It is not feasible to create a Venn diagram for the two independent omics datasets, namely, genes and metabolic substances. These are two sets of independent results, represented by distinct metabolite IDs and gene IDs. Since there is no intersection between them, constructing a Venn diagram is not possible. To improve the clarity, a new figure was constructed. The majority of the DEGs and metabolites were selected and are presented in Figure 5A to improve the clarity of the information. The details of the original high-resolution figure are provided in the revised version as supplementary data. I hope this is satisfactory.
In addition, we revised several minor mistakes.
Round 2
Reviewer 2 Report
Comments and Suggestions for Authors
This manuscript has been well developed, I think it should be published.
Author Response
Dear Reviewer,
Thank you for your positive feedback and acceptance of our manuscript. We sincerely appreciate the time and effort you have dedicated to reviewing our work. Your insightful comments have significantly contributed to improving the quality of this paper.
Best regards,
Peng Luo
South China Sea Institute of Oceanology, CAS
Reviewer 3 Report
Comments and Suggestions for Authors
The authors have addressed my concerns. I don't have more questions now.
Author Response
Dear Reviewer,
Thank you for your positive feedback and acceptance of our manuscript. We sincerely appreciate the time and effort you have dedicated to reviewing our work. Your insightful comments have significantly contributed to improving the quality of this paper. In addition, we further improved the whole article again to make the manuscript more perfect.
Best regards,
Peng Luo,
South China Sea Institute of Oceanology, CAS